# Insight into the Economic Effects of a Severe Korean PRRSV1 Outbreak in a Farrow-to-Nursery Farm

**DOI:** 10.3390/ani12213024

**Published:** 2022-11-03

**Authors:** Jung-Hee Kim, Seung-Chai Kim, Hwan-Ju Kim, Chang-Gi Jeong, Gyeong-Seo Park, Jong-San Choi, Won-Il Kim

**Affiliations:** 1Department of Veterinary Clinic, Dodram Pig Farmers Cooperative, Daejeon 35377, Korea; 2Department of Veterinary Medicine, Jeonbuk National University, Iksan 54596, Korea; 3Department of Agri-Food Marketing, Jeonbuk National Univeristy, Jeonju 54896, Korea

**Keywords:** porcine reproductive and respiratory syndrome virus, economic impact, production impact

## Abstract

**Simple Summary:**

The impairment of reproductive and growth performance of pigs by disease is directly connected to the economic loss of a farm. Porcine Reproductive and Respiratory Syndrome (PRRS) is one of the diseases with the highest economic impact prevailing globally. However, the impact of PRRS outbreaks on swine production in South Korea has been rarely reported. By monitoring before and after a PRRS outbreak in a PRRS-negative farm with mass vaccination in sows and gilts, we found that a potential vaccine failure caused by PRRSV infection in this farm caused severe losses in both the farrowing period and growing period, which is similar to or exceeds reports from other countries. As a result, the pig farm in this study suffered extensive production and economic losses due to PRRSV infection, and further studies are needed to estimate the total cost of economic losses due to PRRS outbreaks in the South Korean swine industry.

**Abstract:**

Porcine reproductive and respiratory syndrome (PRRS) is a disease that has inflicted economic losses in the swine industry. The causative agent, porcine reproductive and respiratory syndrome virus (PRRSV), is known to have a high genetic diversity which leads to heterogeneous pathogenicity. To date, the impact of PRRS outbreaks on swine production and the economy of the swine industry in South Korea has been rarely reported. In this study, we compare the reproductive performance in the breeding-farrowing phase and growth performance in the nursery phase, in two 27-week periods, one before and one after a PRRSV1 outbreak on a 650-sow farrow-to-nursery farm caused by a Korean PRRSV1 isolate which was genetically distinct from vaccine strains or other global strains. The reproductive performance of sows and the growth performance of nursery pigs were compared using row data consisting of 1907 mating records, 1648 farrowing records, and 17,129 weaning records from 32 breeding batches. The following variables were significantly different between the pre-PRRS outbreak period and the post-PRRS outbreak period: the farrowing rate (−7.1%, *p* < 0.0001), the abortion rate (+3.9%, *p* < 0.0001), the return rate (+2.9%, *p* = 0.0250), weaning to estrus interval days (+1.9 days, *p* < 0.0001), total piglets born (−1.2 pigs/litter, *p* < 0.0001), piglets born alive (−2.2 pigs/litter, *p* < 0.0001), weaned piglets (−2.7 pigs/litter, *p* < 0.0001), pre-weaning mortality (+7.4%, *p* < 0.0001), weaning weight (−0.9 kg/pig, *p* = 0.0015), the mortality rate (+2.8%, *p* < 0.0001), average daily gain (−69.8 g/d, *p* < 0.0001), and the feed conversion ratio (+0.26, *p* = 0.0036). Economic losses for a period of 27 weeks after a PRRS outbreak were calculated at KRW 99,378 (USD 82.8) per mated female for the breeding-farrowing phase, KRW 8,968 (USD 7.5) per pig for the nursery growth phase, and KRW 245,174 (USD 204.3) per sow in the post-outbreak period. In conclusion, the farrow-to-nursery farm in our study suffered extensive production and economic losses as a result of a PRRSV1 outbreak.

## 1. Introduction

Porcine reproductive and respiratory syndrome (PRRS), initially recognized in the United States in 1987 [1], is one of the major health challenges in the global swine industry [2,3]. The causative agent, PRRS virus (PRRSV), belonging to the genus *Porarterivirus* in the family *Arteriviridae*, is an enveloped, single-stranded positive-sense RNA virus with a genome approximately 15 kb in length [4]. Based on sequence analysis by the International Committee on Taxonomy of Viruses (ICTV), the genotypes of PRRSV, PRRSV1 (European genotype) and PRRSV2 (North American genotype), are classified into two distinct viral species as *Betaarterivirus suid* 1 and *Betaarterivirus suid* 2 which shares only 60% of nucleotide similarity at the whole genome level [5].

In South Korea, PRRSV2 was first identified in 1994, while PRRSV1 has spread rapidly since its first detection in 2005 [6,7]. A nation-wide vaccination procedure has been conducted to control the disease since PRRSV2-modified live vaccine (MLV) was first licensed in 1996 (Ingelvac^®^ PRRS MLV), followed by an additional PRRSV2 MLV (Fostera^®^ PRRS) and PRRSV1 MLVs (Porcilis^®^ PRRS and UNISTRAIN^®^ PRRS MLV), which were commercialized in 2014 [8,9]. However, epidemiological studies suggested that both PRRSV1 and PRRSV2 are highly prevalent in Korean herds, and some herds are infected with both species [10,11,12,13]. Korean PRRSV1 has been reported to be divided into subgroups A (sub1A), B (sub1B), and C (sub1C) of subtype 1 (pan-European PRRSV1), in which the majority are PRRSV1 sub1A while the vaccine-like sub1C comprises only a minor population [10,11]. On top of the high prevalence of sub1A viruses, PRRSV1 MLV vaccination in Korean PRRSV1-positive (sub1A) farms showed an increased viral genetic heterogeneity when comparing the viral genetics before and after vaccine adoption, suggesting an enhanced diversifying evolution and immune evasion of Korean PRRSV1 sub1A field isolates against commercial PRRSV1 MLVs (sub1C) [9,14].

Several studies attempted to calculate the observed economic impact of a PRRS outbreak at the individual farm level. An initial Dutch study estimated the economic loss of PRRS outbreaks at around EUR 65 per sow per year in 91 breeding and/or farrow-to-finish herds [15]. In the first 18 weeks after a PRRS outbreak in the Netherlands, the economic losses varied between EUR 59 and EUR 379, and the mean loss per sow was EUR 126 [16]. Dee, Joo, Polson, and Marsh reported that increased mortality rates, medication costs, and reduced growth rates caused by a PRRS outbreak exacted mean economic costs of USD 228 per sow for one year [17]. When the farrow-to-finish farm (1000 sows) was severely affected in all stages, annual median losses were EUR 650,090 and overall losses were slightly higher in the breeding phase compared to nursery and fattening phases [18]. It is difficult to compare studies of production losses resulting from the introduction of PRRSV due to differences in the virulence of circulating viruses, production systems, and the calculating system of the losses [18,19]. Generally, a single infection of PRRSV1 is known to show less pathogenicity than a single infection of PRRSV2 or co-infection of PRRSV1 and PRRSV2 together [20]. However, the emergence of highly pathogenic PRRSV1 subtype 3 strains in Eastern Europe and reports of PRRSV1 recombinant strains from several countries pose a potential threat of PRRSV1 outbreaks with severe economic losses [21,22,23,24,25,26,27]. Indeed, a recent Danish study reported high production losses after a recombinant PRRSV1 strain outbreak with lower farrowing rates per week (0.1–10.8%), less liveborn piglets per litter (0.8–4.8 pigs), and less weaned piglets per litter (2.4–6.5 pigs), of which the losses exceeded the losses normally seen in not only Danish PRRSV-infected herds but also other regions and countries [19].

Unfortunately, production and economic losses attributed to PRRSV1 and/or PRRRSV2 at an individual farm level is rarely reported in South Korea. In a 650-sow farrow-to-nursery farm producing PRRSV-seronegative pigs from weaning to 80 days of age, we confirmed PRRSV1 in aborted stillborn fetuses and had collected data until the farm reached PRRSV-stability. The aim of this study was to estimate the production and economic losses in a 27-week period after a severe case of a Korean PRRSV1 outbreak in a farrow-to-nursery farm in Korea.

## 2. Materials and Methods

### 2.1. Study Population and PRRSV Outbreak History

The study was performed on a commercial 650-sow farrow-to-nursery pig farm in Sunchang county, a southern region in South Korea. The farm is located at the following geographical coordinates (latitude/longitude): 35°26′32.2″ N 127°10′03.0″ E. Biosecurity procedures were carried out rigorously according to veterinarian instructions. The farm generally weaned a batch of 625 pigs every 12 days and had 3 separate farrowing rooms. Weaning piglets at 24 days of age were housed in 6 separate sites with 8 barns per site for 7 weeks. Each nursery site consisted of fully slatted-floors and a negative-pressure system, and a stocking density of 0.3 m^2^ per pig. The nursery sites were kept at a constant temperature between 25 and 30 °C depending on body weight. The farm produced piglets in a 12-day batch farrowing interval and the suckling period was about 24 days. At the end of the nursery period, 25–30 kg piglets were transferred to the fattening unit located 50 km from the breeding herd or sold to other finishing farms.

Prior to the PRRS outbreak, the productive performance at this farm was superior to those recorded at other farms in South Korea. In 2018, the performance was as follows: 89.3% (farrowing rate), 14.0 pigs (total born per litter), 13.1 pigs (born-alive per litter), 11.7 pigs (weaned per litter), and 28.9 pigs (weaned per sow per year). Since 2014, this farm has produced seronegative weaned piglets remaining PRRSV-negative until they became nursery pigs at 70 days of age, based on routine serological testing. As a result of vaccination, sows and gilts were seropositive for PRRSV. Mass vaccination was applied to sows twice a year and to gilts twice at a 4-week interval in the quarantine facility using a modified live PRRSV1 vaccine UNISTRAIN^®^ PRRS (Hipra, Girona, Spain).

PRRSV infection on this farm was confirmed on 2 April 2019 (29th breeding batch–mated females during 5 December to 12 December 2018), with clinical problems starting at the end of March 2019. There was a marked reduction in feed intake and high fevers (>40 °C) in 6 pregnant sows and about 30 sows aborted at 80–110 days of gestation within a 3-week period. Mass vaccination for farm immunization was applied to breeding herds twice at a 4-week interval using a UNISTRAIN^®^ PRRS vaccine.

### 2.2. Genetic Characteristics of PRRSV

Clinical samples, including serum and lungs, were collected from aborted stillborn fetuses and submitted to Jeonbuk National University Diagnostic Center (JBNC-VDC). A causative PRRSV1 was isolated from lung homogenate (10% [weight/volume]) of finely chopped lung pieces prepared in Dulbecco-modified Eagle medium (DMEM) by inoculating into primary cultures of porcine alveolar macrophages (PAMs) which was collected from a 4-week-old piglet and maintained in our lab as previously described [28,29]. The PRRSV1 isolate was named the JBNU-19-E01 strain. The whole genome of the isolate was sequenced from the supernatant of the virus culture (second passage) in PAM through next-generation sequencing (NGS) using the Illumina iSeq100 platform with 150 bp paired-end read and the whole-genome sequence was gathered with an in-house analysis pipeline [30]. The obtained whole-genome sequence was submitted to Genbank under accession number MW847781. Phylogenetic analysis constructing a maximum likelihood tree was conducted together with the PRRSV1 whole genome and ORF5 reference sequences using MEGA software [31]. The JBNU-19-E01 strain was found to be classified into the Korean subtype 1A according to ORF5 lineage classification and at the whole-genome level (Figure 1). Differences in nucleotide sequence homology between the JBNU-19-E01 strain and other Korean subtype 1A viruses varied in the range of 8.9~12.5% at the whole-genome level (Appendix A). No sign of genomic recombination of the JBNU-19-E01 strain with other subtype 1A viruses or MLVs was detected through the Recombination Detection Program (RDP) [32] and Simplot analysis [33].

### 2.3. Assessment of the Production and Economic Impact

To evaluate the production and economic losses from a Korean PRRSV1 outbreak, the periods were marked out for 27 weeks (16 batches) before and 27 weeks (16 batches) after the PRRS outbreak based on the last time weaning piglets and nursery pigs were RT-qPCR positive, which was in the 14th breeding batch (females were mated between 31 May to 12 June 2019) (Figure 2).

Data on reproductive and growth performance were collected from the software program Pig Plan^®^ software (EZ Farm Inc., Anyang, South Korea). The data dictionary was based on traditional definitions of industry terms and formulae. Reproductive performance data on the gilts and sows were collected from breeding records from May 2018 to June 2019 and farrowing records from September 2018 to October 2019. The collected data included sow identities, breeding dates and results, parity, gestation and lactation length, weaning to estrus intervals (WEI), the percentage of sows mated within 7 days after weaning (%), farrowing rates (FR), abortion rates (AR), return rates (RR), total piglets born per litter (TB), piglets born alive per litter (BA), stillborn piglets per litter (SB), mummified fetuses per litter (MM), and weaned piglets per litter (WP). Growth performance data on the nursery pigs were collected from weaning records from September 2018 to October 2019 including pre- and post-weaning mortality rates (%), body weight at weaning, 55 days of age, and 77 days of age (kg), average daily gain (ADG), average feed intake (AFI), and feed conversion rates (FCR). The raw data pool consisted of 1907 mating sows, 1648 farrowing sows and 17,129 weaned piglets from 32 breeding batches.

The negative economic impact of a Korean PRRSV1 outbreak has been assessed for two different production phases: breeding-farrowing and nursery. For the breeding-farrowing phase, the calculation has been conducted by examining the reduced revenue (in terms of weaned piglets per mated female) and increased costs (additional feed and insemination costs) from a failure to conceive or farrow and by delayed the weaning-to-estrus interval per mated female. The average feed intake per mated female was 4.0 kg/day (from weaning to first service) and 2.5 kg/day (from first service to pregnancy failure or abortion). Feed and two-dose insemination costs per mated female were KRW 425 (USD 0.4)/kg and KRW 14,000 (USD 11.7), respectively. The economic impact per pig in the nursery phase was evaluated by reduced weaning weight, feed efficiency, average daily gain, and increased mortality. The facility cost of the nursery phase was calculated as USD 0.10 per pig per day [2]. Average feed cost during the nursery period was KRW 430 (USD 0.4). The average price of piglets at weaning was calculated as KRW 5053 (USD 4.2)/kg based on 2972 nursery pigs (average 28.8 kg) sold in 2019. On the evidence of business analysis of this farm in 2020, the fixed costs (capital investment in building and equipment, insurance, repairs, taxes, and interest) and non-feed variable costs (animal health, labor, and farm management) per weaned pig were considered.

### 2.4. Statistical Analysis

Statistical analyses were conducted using GraphPad Prism version 8.0 for Windows (GraphPad Software, San Diego, CA, USA). The data were analyzed for differences over time, i.e., 27 weeks before PRRSV field infection (16 batches), and 27 weeks after PRRSV field infection (16 batches). The Shapiro–Wilk test was used to determine whether random samples had a normal distribution. The average number of mated and farrowed females, body weight, ADG, AFI and FCR per batch were compared using the Unpaired *t*-test. Gestation and lactation length, parity, WEI, TB, BA, SB, MM, and WP showed a non-normal distribution, and were thus compared with the Mann–Whitney test. Pre- and post-weaning mortality, return, abortion, farrowing, and early parturition (<114 days) rates were compared with the Fisher’s exact test. *p*-values < 0.05, <0.001 and <0.0001 were considered to be statistically significant, very significant and extremely significant, respectively.

## 3. Results

### 3.1. Production Impact of a PRRS Outbreak in the Breeding-Farrowing and Nursery Phases

Reproduction performance (FR, RR, AR, WEI, TB, BA, SB, MM, and WP) over time is summarized in Table 1. During the 27 weeks after a PRRS outbreak, reproductive performance had a statistically significant difference when compared with pre-outbreak performance: FR (82.9 vs. 90.0 %, *p* < 0.0001), AR (5.0 vs. 1.2 %, *p* < 0.0001), RR (10.4 vs. 7.5, *p* = 0.0250) (Table 1 and Figure 3), WEI (6.5 vs. 4.6 days, *p* < 0.0001), the percentage of sows mated within 7 days after weaning (88.0 vs. 96.5%, *p* < 0.0001), early parturition rates (<114 days; 22.3 vs. 6.5%, *p* < 0.0001), TB (13.3 vs. 14.5 piglets/litter, *p* < 0.0001), BA (11.4 vs. 13.6 piglets/litter, *p* < 0.0001), WP (9.0 vs. 11.7 piglets/litter, *p* < 0.0001) (Table 1 and Figure 4),SB (1.0 vs. 0.6 piglets/litter, *p* < 0.0001), and MM (0.9 vs. 0.3 piglets/litter, *p* < 0.0001) (Table 1 and Figure 5).

Growth performance in the nursery phase (pre- and post-weaning mortality, ADG, AFI, FCR, and body weight at weaning, 55 days of age, and 77 days of age) over time is summarized in Table 2 and Figure 6. During the 27 weeks after the PRRS outbreak, pre-weaning mortality, post-weaning mortality from 24 to 53 days of age, and from 24 to 77 days of age increased by 7.4%, 0.9%, and 2.8%, respectively (*p* < 0.0001). The average body weight at weaning, 55 days of age, and 77 days of age decreased to 0.9 kg (*p* = 0.0015), 2.6 kg (*p* < 0.0001), and 4.2 kg (*p* < 0.0001), respectively. The average daily gain from 24 to 53 days of age, and from 24 to 77 days of age decreased by 49.5 g/d (*p* = 0.0002) and 69.8 g/d (*p* < 0.0001), respectively. From 24 to 53 days of age, the average feed intake decreased by 51.5 g/d (*p* = 0.0795) and the feed conversion ratio increased by 0.26 (*p* = 0.0036).

### 3.2. Economic Impact of the PRRS Outbreak for the Breeding-Farrowing Phase on Gilts and Sows

The economic losses in the breeding-farrowing phase were estimated to be KRW 99,378 (USD 82.8) per mated female (Table 3). We determined that the cost of a weaned pig was KRW 29,449 (USD 24.5) after the PRRS outbreak based on 5.83 kg/pig at weaning. Thus, there was a reduction in revenue of KRW 90,860 (USD 75.7) per mated female owing to a drop in the number of pigs weaned per mated female (10.5 vs. 7.4). Additional feed consumption resulting from delayed weaning to estrus intervals was 6.37 kg per mated female (except for gilts accounting for 14.1% of every breeding batch). Moreover, additional feed consumption caused by increased return-to-estrus and abortion rates was 2.14 kg and 9.29 kg per mated female. The calculation methods of feed consumption were as follows: the number of gilts and sows returned to estrus after first service was 0.07 (pre-outbreak) and 0.10 (post-outbreak) per mated female, and aborted after first service was 0.01 (pre-outbreak) and 0.05 (post-outbreak) per mated female; the average outbreak period from first service to return-to-estrus was 25.5 (pre-outbreak) and 26.6 (post-outbreak) days, and from first service to abortion was 50.7 (pre-outbreak) and 86.0 (post-outbreak) days. Thus, additional feed costs were KRW 7568 (USD 6.3) per mated female. The additional insemination costs caused by increased return-to-estrus and abortion rates was KRW 950 (USD 0.8) per mated female.

### 3.3. Economic Impact of the PRRS Outbreak for the Growth Phase on Nursery Pigs

The economic impact of PRRS was also assessed for the nursery phase. Due to an increase in mortality, a reduction in weaning weight, feed efficiency, and an average daily gain, the economic loss in the nursery phase was calculated at KRW 5973 (USD5.0) per pig (Table 4). The increase in mortality from 24 to 77 days of age was 2.83%. Considering that a weaned pig cost KRW 29,449 (USD 24.5), the increased cost was KRW 857 (USD 0.7) per pig for the nursery growth phase. The total feed intake required to reach 30 kg after weaning was 47.0 (pre-outbreak) and 55.2 kg (post-outbreak) per pig owing to the reduced weaning weight (−0.9 kg) and the increased feed conversion ratio (12.9%). Thus, the increased feed costs during the nursery phase was KRW 3497 (USD2.9) per pig. The pig-rearing period required to reach 30 kg after weaning was 55.2 (pre-outbreak) and 68.6 (post-outbreak) days owing to the reduced ADG (16.5%). With a facility cost of USD 0.10/pig/day [2], the increased facility cost was KRW 1620 (USD 1.3)/pig. Increased fixed and non-feed variable costs were KRW 2995 (USD 2.5). In summary, the economic loss in the nursery phase was KRW 8968 (USD 7.5) per pig.

## 4. Discussion

The production losses and economic impacts in a commercial 650-sow farrow-to-nursery pig farm infected with Korean PRRSV1 field strain has been documented in the present study. The reproductive performance of sows and the growth performance of nursery pigs were compared using row data consisting of 1907 mating records, 1648 farrowing records, and 17,129 weaning records from 32 breeding batches between the pre-outbreak period (27 weeks; 16 batches) and the post-outbreak period (27 weeks; 16 batches).

In this study, clinical signs of PRRS included marked feed intake reduction, high fever, and a marked increase in late-term abortions in spite of regular mass vaccination of the sows and gilts with a PRRSV1 MLV. According to the subsequent whole-genome sequencing analysis of the PRRSV isolated from the farm, the JBNU-19-E01 strain was identified to be classified into Korean PRRSV1 subtype 1A. Since PRRSV1 was first reported in Korea in 2005, the PRRSV1 sub1A group has been the major PRRSV1 population comprising more than 90% of PRRSV1 infection cases in Korea [14]. Although commercial PRRSV1 MLV vaccines were first commercialized in Korea in 2014, these MLV vaccines (sub1C group) are known to show genetic heterogeneity with Korean PRRSV1 sub1A group (Figure 1 and Appendix A) [9,10,14]. In addition, a previous study highlighted different deletion patterns in the nonstructural protein 2 (NSP2) region of Korean PRRSV1 subtype 1A compared to vaccine-like sub1C, implying independent evolution of subtype 1A viruses in the field [30]. PRRSV commercial vaccines often fail to confer sterilizing immunity against field strains [9]. They are generally known to offer better protection efficacy against wild viral variants that have a higher degree of similarity to the original vaccine parental isolate [34,35]. Thus, the PRRSV1 outbreak in this study could be explained as a first case report of potential PRRSV1 MLV vaccine failure in Korea as there has been limited information regarding Korean PRRSV1, especially after the introduction of the vaccines in 2014 [9].

We evaluated the time profile of the production impact of PRRS, identifying each change in batch (at a 12-day interval) yield compared with the pre-outbreak baseline. The results showed that the time to PRRSV stability (TTS) was 27.0 weeks and the time to baseline production (TTBP) was as follows: abortion rate (17.9 weeks), prenatal losses (difference between total born and live born piglets/litter, 25.1 weeks), pre-weaning mortality (19.7 weeks), weaned piglets per litter (33.4 weeks), body weight (28.1 weeks at weaning and 25.1 weeks at 77 days of age), and post-weaning mortality (17.9 weeks from 24 to 77 days of age). The PRRSV1 outbreak in this study caused a 7.1% decline in the farrowing rate and a 16.0% (2.2 piglets) decrease in piglets born alive per litter in the 27 weeks after the outbreak. Correspondingly, PRRS reduced production by 3.09 in weaned pigs per mated female. This decrease is substantially larger than the reduction of 1.44 weaned pigs per mated female in a previously reported study [3]. The increase in abortions was the most powerful signal of a PRRS outbreak in this study. Abortions rose two weeks after a marked reduction in feed intake. The abortion rates in the third and fourth batches after the PRRS outbreak were 16.9% and 21.0% (Figure 3). In our study, the average weaned piglets per litter remained below the baseline throughout the 33.4 post-PRRS outbreak weeks. Pre-weaning mortality increased rapidly in four batches shortly after the outbreak. The pre-weaning mortality rates for these batches were 44.2%, 47.1%, 25.7%, and 40.4% (Figure 5). Stillbirth rates remained elevated at four batches shortly after the outbreak, suggesting that infected sows produced 1.1 to 3.0 dead piglets per litter.

Between the 32 batches, the economic impact of PRRS in the breeding-farrowing phase was calculated to be KRW 99,378 (USD 82.8) per mated female up to a total of KRW 95,303,561 (USD 79,420) across the 27 weeks post-outbreak. The costs incurred from PRRS in the nursery growth phase were estimated to be KRW 8968 (USD 7.5) per pig and KRW 64,059,300 (USD 53,383) over the 27 weeks post-outbreak. The economic losses for this farrow-to-nursery farm (650 sows) were estimated to be KRW 245,174 (USD 204.3) per sow over the 27 weeks post-outbreak.

Comparing economic losses attributed to PRRS varies between the PRRS virus strain, immunity, management, pig density, and the housing system. As different countries have different ways of calculating and reporting data, it is even more difficult to make direct comparisons with other studies of production losses resulting from the introduction of PRRSV in different regions/countries [19]. In previous studies, a reduction in liveborns per litter of between one and two pigs have been reported after PRRSV infection [3,16,36,37,38,39]. Similarly, an increased preweaning mortality from 4 to 17% [3,15,16,36,37,38] and a decreased farrowing rate [2,40,41] have been reported in outbreaks in other countries. As a result of these production losses due to a decline in farrowing and increased piglet mortality, the number of weaned pigs per litter decreased by two to three pigs [2,36,39]. In this study, it is arguable whether or not the one farm investigated in this study generates enough information to represent PRRSV1 infections in general pig farms in South Korea. However, it is clear that the losses seen in connection with the PRRSV1 infection in this study was similar to or exceeded the losses seen in connection with PRRS outbreaks in other regions and countries. Further studies are needed to estimate the total cost of economic losses due to PRRS outbreaks in the South Korean swine industry.

## 5. Conclusions

In the present study, we analyzed the economic and production impacts before and after a Korean PRRSV1 outbreak on a farrow-to-nursery farm with mass vaccination in sows and gilts. TTS and TTBP (for weaned piglets per litter) were 27.0 and 33.4 weeks, respectively. PRRS imposed tremendous economic damage and cost approximately KRW 245,174 (USD 204.3) per sow during a period of 27 weeks after a PRRS outbreak by reducing the reproductive performance of sows and the growth performance of nursery pigs. Based on this assessment, it can be concluded that the impact of potential PRRSV1 vaccine failure caused by Korean subtype 1A virus was similar to or exceeded that reported in other countries.

## Figures and Tables

**Figure 1 animals-12-03024-f001:**
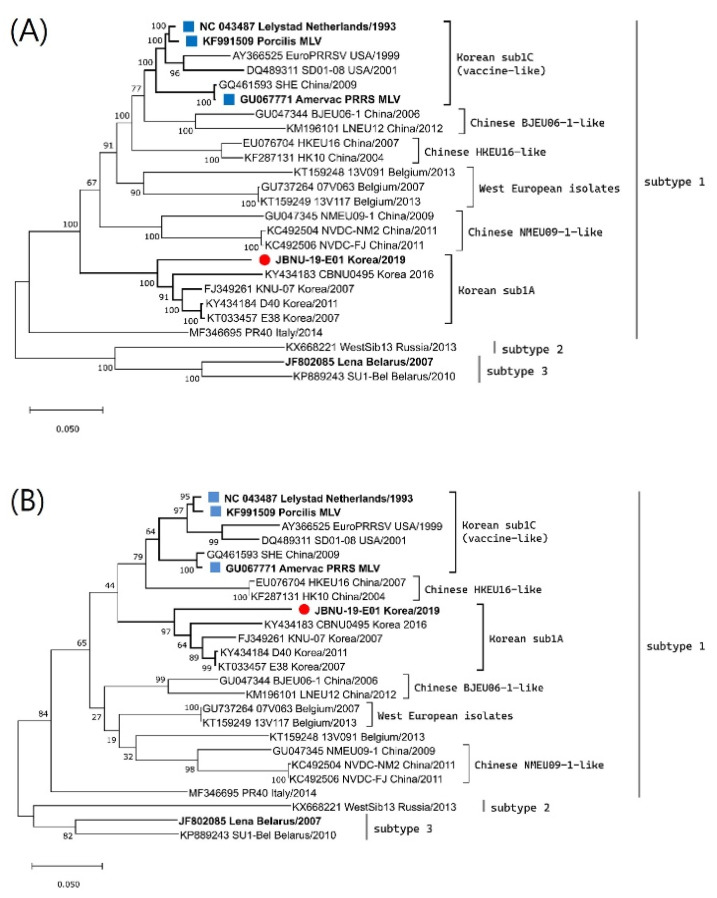
Phylogenetic trees of PRRSV1 (**A**) whole genome and (**B**) ORF5 sequences using the maximum likelihood method based on the generalized time-reversible (GTR) model with G + I in MEGA X. Bootstrap values were calculated with 1000 replicates. The isolate newly sequenced in this study was marked with a red circle, while the prototype (Lelystad) and vaccines were marked with a blue square.

**Figure 2 animals-12-03024-f002:**
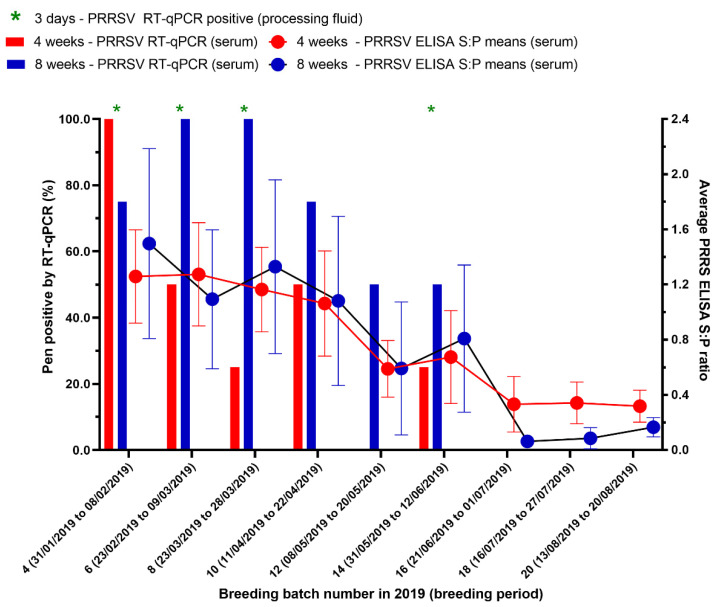
The results of testing processing fluids and serum by real-time reverse transcription-polymerase chain reaction (RT-qPCR) and testing serum by enzyme-linked immunosorbent assay (ELISA) for PRRSV in pigs. Ct-value ≤ 35 and an S/P ratio ≥ 0.4 were considered to be positive. Breeding batches that are PRRSV RT-qPCR positive from processing fluid are indicated (*).

**Figure 3 animals-12-03024-f003:**
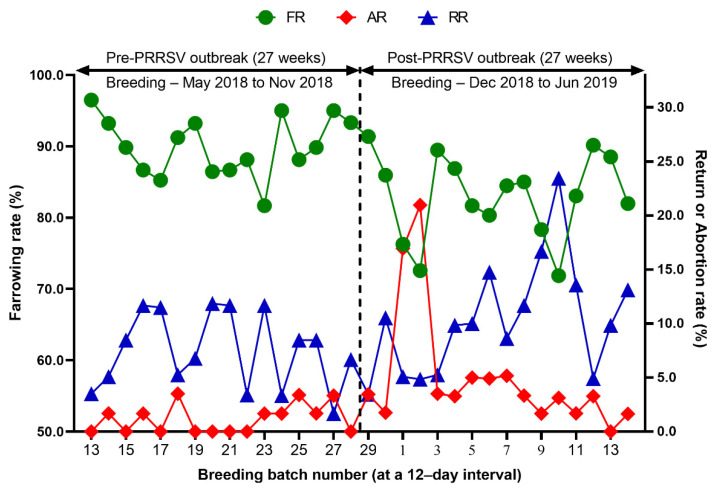
Comparison of the farrowing rate (FR), the abortion rate (AR), and the return rate (RR) before and after a PRRS outbreak.

**Figure 4 animals-12-03024-f004:**
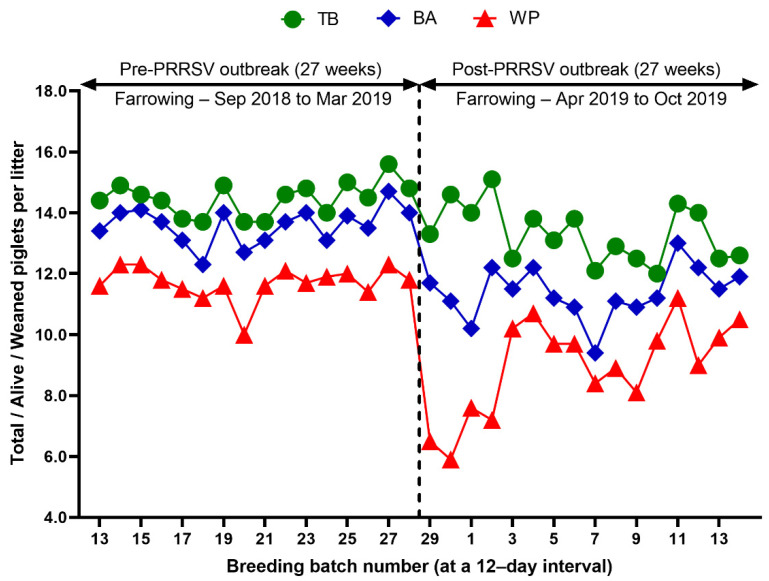
Comparison of total piglets born per litter (TB), piglets born alive per litter (BA), and weaned piglets per litter (WP) before and after a PRRS outbreak.

**Figure 5 animals-12-03024-f005:**
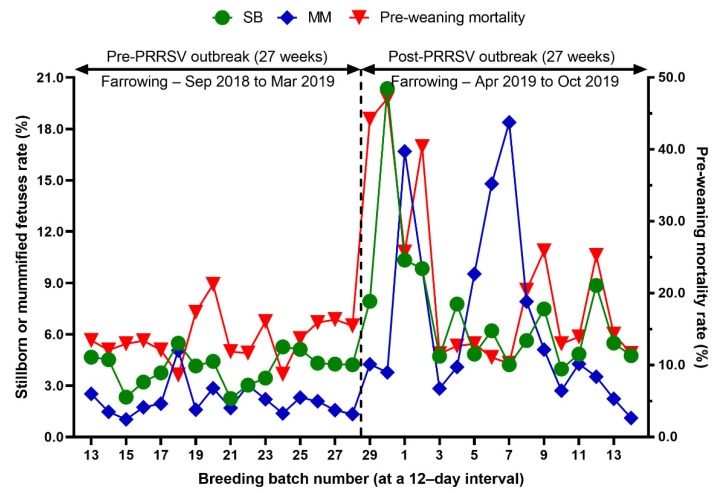
Comparison of the percentage of stillborn piglets (SB) and mummified fetuses (MM), pre-weaning mortality rate before and after a PRRS outbreak.

**Figure 6 animals-12-03024-f006:**
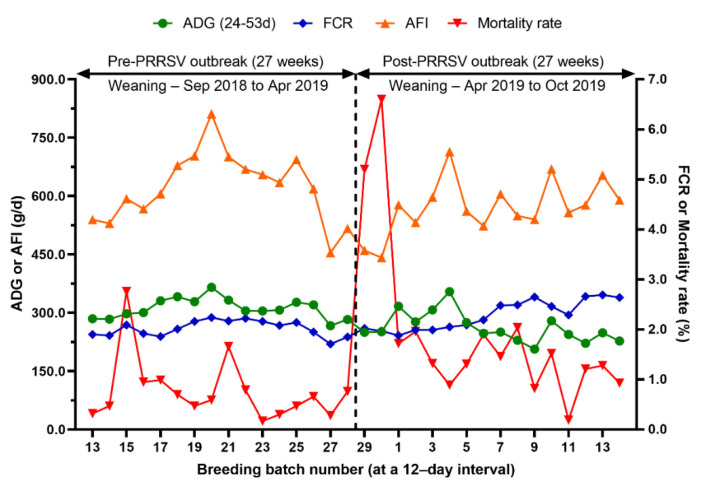
Comparison of average daily gain (ADG), feed conversion ratio (FCR), average feed intake (AFI) and mortality rate from 24 to 53 days of age before and after a PRRS outbreak.

**Table 1 animals-12-03024-t001:** Comparison of the reproductive performance of gilts and sows before and after a porcine reproductive and respiratory syndrome (PRRS) outbreak on a farrow-to-nursery farm.

Reproductive Performance	Pre-Outbreak	Post-Outbreak	Difference	*p*-Value
Number of matings (average/batch)	948 (59.3)	959 (59.9)	11	0.2126
Number of farrowings (average/batch)	853 (53.3)	795 (49.7)	−58	0.0006
Parity	4.0	4.2	0.2	0.1063
Gestation length (days)	115.1 ± 1.2	115.3 ± 1.5	0.2	<0.0001
Early parturition rate (<114 days, %)	6.5	22.3	15.8	<0.0001
Farrowing rate (%) (FR)	90.0	82.9	−7.1	<0.0001
Abortion rate (%(AR)	1.2	5.0	3.9	<0.0001
Return rate (%) (RR)	7.5	10.4	2.9	0.0250
Weaning to estrus intervals (days) (WEI)	4.6 ± 2.3	6.5 ± 6.3	1.9	<0.0001
The percentage of sows mated within 7 days after weaning (%)	96.5	88.0	−8.5	<0.0001
Total piglets born per litter (TB)	14.5 ± 4.0	13.3 ± 4.4	−1.2	<0.0001
Piglets born alive per litter (BA)	13.6 ± 3.7	11.4 ± 4.2	−2.2	<0.0001
Stillborn piglets per litter (SB)	0.6 ± 1.0	1.0 ± 1.9	0.4	<0.0001
Mummified fetuses per litter (MM)	0.3 ± 0.7	0.9 ± 2.2	0.6	<0.0001
Weaned piglets per litter (WP)	11.7 ± 2.0	9.0 ± 3.5	−2.7	<0.0001
Lactation length (days)	24.0 ± 2.4	24.0 ± 2.3	0.0	0.4077

**Table 2 animals-12-03024-t002:** Comparison of the growth performance of nursery pigs before and after a porcine reproductive and respiratory syndrome (PRRS) outbreak on a farrow-to-nursery farm.

Growth Performance	Pre-Outbreak	Post-Outbreak	Difference	*p*-Value
Pre-weaning mortality (%)	13.8	21.3	7.4	<0.0001
Body weight at weaning (kg)	6.7 ± 0.7	5.8 ± 0.8	−0.9	0.0015
Nursery period (24–53 days)				
Body weight at 53d (kg)	16.3 ± 1.5	13.7 ± 1.8	−2.6	<0.0001
ADG (g/d)	311.2 ± 25.8	261.7 ± 38.6	−49.5	0.0002
AFI (g/d)	623.3 ± 89.1	571.8 ± 70.2	−51.5	0.0795
FCR	2.02 ± 0.16	2.28 ± 0.29	0.26	0.0036
Mortality rate (%)	0.8	1.7	0.9	<0.0001
Nursery period (24–77 days)				
Body weight at 77d (kg)	29.5 ± 1.0	25.3 ± 1.9	−4.2	<0.0001
ADG (g/d)	421.9 ± 19.5	352.1 ± 31.0	−69.8	<0.0001
Mortality rate (%)	2.1	4.9	2.8	<0.0001

**Table 3 animals-12-03024-t003:** The economic impact of PRRS on the breeding-farrowing production phase.

Variable	Value
Reduced number of weaned pigs/mated female	
Difference in weaned pigs/mated female	3.1
Value of weaned pig/kg	KRW 5053 (USD 4.2)
Reduced revenue/mated female	KRW 90,860 (USD 75.7)
Additional feed consumption/mated female	
Feed price/kg	KRW 425 (USD0.4)
Feed intake from weaning to first service	4.0 kg/day
Feed intake from first service to return-to-estrus or abortion	2.5 kg/day
Delayed weaning to estrus intervals	1.9 days
Increased feed cost/mated female	KRW 7568 (USD6.3)
Wasted semen/mated female	
Increased insemination cost/mated female	KRW 950 (USD 0.8)
Total cost/mated female	KRW 99,378 (USD 82.8)

USD 1 = KRW 1200.

**Table 4 animals-12-03024-t004:** The economic impact of PRRS on the nursery growth phase.

Variable	Value
Cost of increased mortality rate	
Increased death loss	2.83%
Value of a weaned pig	KRW 29,449 (USD 24.5)
Cost of death loss/pig	KRW 833 (USD 0.69)
Adjusted cost of death loss	KRW 857 ^†^ (USD 0.71)
Cost of reduced weaning weight and feed efficiency	
Feed price/kg	KRW 430 (USD 0.4)
Difference in total feed intake to reach 30kg/pig	8.13 kg
Increased feed cost/pig	KRW 3497 (USD 2.9)
Cost of reduced weaning weight and average daily gain	
Facility cost/day [2]	KRW 120 (USD 0.1)
Difference in pig-rearing period to reach 30kg/pig	13.5 days
Increased facility cost/pig	KRW 1620 (USD 1.3)
Increased fixed and non-feed variable costs	
Labor cost/pig	KRW 1061 (USD 0.9)
Building and facility cost/pig	KRW 458 (USD 0.4)
Insurance and tax	KRW 256 (USD 0.2)
Water, power, fuel and transport cost/pig	KRW 298 (USD 0.2)
Veterinary and medicine cost/pig	KRW 314 (USD 0.3)
Excretion disposal cost/pig	KRW 517 (USD 0.4)
Interest and miscellaneous materials cost/pig	KRW 91 (USD 0.1)
Total increased cost/pig	KRW 8968 (USD 7.5)

USD 1 = KRW 1200, ^†^ Cost per pig divided by the percentage survivability in the nursery phase of production (KRW 833/0.972).

## Data Availability

The raw data including the present data can be shared because we could obtain the necessary permissions from the relevant herd owner.

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
