# Peer review of "Insight into the Economic Effects of a Severe Korean PRRSV1 Outbreak in a Farrow-to-Nursery Farm"

_animals, 2022, doi:10.3390/ani12213024_

Round 1

Reviewer 1 Report

A detailed look at economic data of swine production prior at in the immediate aftermath if infection in the face of vaccination pressure. This is important to review periodically to guide research investments and evaluate vaccine efficacy in a meaningful economic way. 

The others do a fine job of collating and presenting production and economic data from a large swine farm preceding and following infection with PRRS virus. This illustrates the cost of these viral infections in very real terms and make evident the need for better control strategies.

The strength of the study is the very good farm records and the fortuitous opportunity to perform the analysis to present it publicly. The weakness, it is only one outbreak on one farm. This is addressed somewhat by the reference to past similar papers and the similar impact reported by those authors.

It was a straight forward comprehensive presentation of the data that was being collected on the farm. They could go into a bit more detail on pig flow, ventilation, barn temperatures and biosecurity measures that might impact the disease. I am not sure it is necessary but may be of interest to some readers.

Reviewer 2 Report

This manuscript addresses an important area in PRRS control strategies, documenting the economic impact of infection with PRRSV-1 in South Korea. While specific to the single farm investigated, a farrow-to-nursery system, and therefore limited in overall application, the results presented provide a significant addition to the available data.

A review of the English language used throughout would be beneficial to increase understandability of the manuscript.

Specific comments:

Simple summary, line 2 - Should be Porcine Reproductive....

Abstract, lines 2-3 - PRRSV should be plural since there are two viruses.

Introduction, lines 5-6 - PRRSV-1 and PRRSV-2 are not genotypes, they are distinct species.

Materials and methods, 2.2 line 3 - Please explain how the virus isolation was carried out, and from which sample.

Materials and methods, 2.2 lines 3-4 - What material was used for the sequencing - if culture material, from which passage?

Discussion, lines 10-12 - Could the clinical signs be included as data in the results section?

Discussion - Please distinguish between PRRSV-1 and PRRSV-2 in the other studies mentioned.

Reviewer 3 Report

General approach is very interesting, as to assess to economic losses of PRRSv, based on a real farm situation. The economic relevance is substantial and requires attention of readers. So this paper definitely highlights a relevant topic. Still the approach has limitations, and these need to clearly addressed, or improved.
